# Viral protein instability enhances host-range evolvability

**Hannah M. Strobel, Elijah K. Horwitz, Justin R. Meyer** *

Division of Biological Sciences, University of California San Diego, La Jolla, California, United States of America

* jrmeyer@ucsd.edu

## Abstract

Viruses are highly evolvable, but what traits endow this property? The high mutation rates of viruses certainly play a role, but factors that act above the genetic code, like protein thermostability, are also expected to contribute. We studied how the thermostability of a model virus, bacteriophage λ, affects its ability to evolve to use a new receptor, a key evolutionary transition that can cause host-range evolution. Using directed evolution and synthetic biology techniques we generated a library of host-recognition protein variants with altered stabilities and then tested their capacity to evolve to use a new receptor. Variants fell within three stability classes: stable, unstable, and catastrophically unstable. The most evolvable were the two unstable variants, whereas seven of eight stable variants were significantly less evolvable, and the two catastrophically unstable variants could not grow. The slowly evolving stable variants were delayed because they required an additional destabilizing mutation. These results are particularly noteworthy because they contradict a widely supported contention that thermostabilizing mutations enhance evolvability of proteins by increasing mutational robustness. Our work suggests that the relationship between thermostability and evolvability is more complex than previously thought, provides evidence for a new molecular model of host-range expansion evolution, and identifies instability as a potential predictor of viral host-range evolution.

## Author summary

Understanding how viruses evolve to infect new hosts is critical for predicting host shifts as well as tuning host-range in phage therapy applications. Yet a mechanistic understanding of the molecular steps required to shift hosts has not been achieved. For this study we examined the evolutionary potential of different strains of a model virus, bacteriophage λ, to gain the ability to use a new receptor, a key step in host shifts. We discovered that λ variants with destabilized host-recognition proteins were more likely to evolve the necessary mutations to use the new receptor than stabilized variants. However, destabilization was only beneficial to a certain point and variants with overly unstable proteins lost all function. These results led us to propose a new molecular model for receptor use evolution in λ; 1) destabilizing mutations evolve that provide protein structural flexibility that allows new protein conformations to form that are able to interact with the new receptors, and 2)

**Data Availability Statement:** All numerical data and sequence reads are available from the dryad digital repository (https://doi.org/10.6076/D1NC78) [51].

**Funding:** HMS was funded by the UC San Diego Cell and Molecular Genetics Institutional Training

Grant, project number 5T32GM007240-41 and JRM was funded by UC Multi-campus Research Programs and Initiatives, Grant Number MRI-19-601184. The funders had no role in study design, data collection and analysis, decision to publish, or preparation of the manuscript.

**Competing interests:** The authors have declared that no competing interests exist.

mutations evolve that alter the binding surface chemical properties to assist interactions with the new receptor. Our work with a model virus-host system, points to the potential use of viral stability as a phenotypic indicator of the capacity for virus host-range evolution.

## Introduction

The evolvability of life is evident in its remarkable diversity of forms and persistence through time. While life may be inherently evolvable, it is thought that evolvability is a malleable trait itself subject to evolutionary tinkering [1]. A number of mechanisms have been proposed to enhance evolvability, centering on the increased capacity to produce novel phenotypes [2]. The most straightforward is evolving a higher mutation rate which allows populations to explore phenotypic variation that results from genetic changes [3]. However, this mechanism is constrained by concomitant increases in mutation load that can slow adaptation [4]. There are other qualities of biological systems that can facilitate phenotypic novelty, although these too can have conflicting effects. This manuscript focuses on one such trait, protein thermostability, which is the propensity to resist misfolding when heated. High thermostability can promote evolvability by buffering protein folding against destabilization from mutations, a property called mutational robustness [5], thus allowing genomes to accumulate more mutations and increasing the chance of uncovering the mutations for an innovation [6,7,8,9,10]. However, high thermostability can also limit evolvability because high heat tolerance is often achieved by increasing conformational rigidity [11,12,13,14,15], precluding the formation of non-native conformers that sometimes support new functions [16,17]. Indeed, the conformational flexibility of unstable proteins can be essential for evolvability by allowing non-native conformers, protein molecules that fold into conformations other than the ground-state conformation, to explore promiscuous functions [18,19]. While the most widely accepted view is that thermostability promotes faster protein evolution, most scholars recognize that the relationship between thermostability and evolvability is more complex and that certain types of proteins may be more or less sensitive to either mechanism.

As the world experiences the second year of the COVID-19 Pandemic, the attention of many scientists has turned to the problem of predicting which viral strains are most likely to emerge. Presumably, more evolvable viruses are more likely to gain the mutations and innovations necessary to shift species. Fortunately, the ease with which viruses evolve under laboratory conditions allows for direct experimental tests of the role thermostability plays in evolvability. Such experiments have been conducted by culturing different viral variants with potentially different evolvabilities in parallel under identical conditions, and quantifying evolvability as the viruses' ability to adapt to imposed challenges. So far, this method has revealed opposite results: When genotypes of phage φ6 with high and low robustness were evolved to cope with heat stress, the more robust variant adapted faster [20]. However, a separate study challenged vesicular stomatitis virus with replicating on a novel cell type and reported the opposite pattern: less robust variants were more evolvable [21]. These findings suggest that while robustness is one mechanism that can enhance viral evolvability, there are other mechanisms associated with low robustness that may override it.

One particular class of viral proteins that are relevant for host-shift evolution are host-recognition proteins [22,23,24], and thus knowing the determinants of their evolvability is particularly important. We used bacteriophage λ as a model system to test how thermostability impacts the evolution of receptor recognition. Through coevolution with *E. coli* in the

laboratory, λ can evolve from a specialist able to bind only the ancestral receptor, LamB, to a generalist by gaining the ability to bind a second receptor, OmpF [25]. The mutations that produced expanded receptor recognition occurred in the *J* gene, which encodes λ's host-recognition protein, and were also shown to reduce λ's thermostability, as measured by the rate of decrease in infectious titer [26]. Additional experiments revealed that genetically homogeneous populations of the evolved generalist produced a subpopulation of phage particles that lost infectivity faster than the population as a whole, and the unstable subpopulation also possessed enhanced ability to use OmpF [26]. This system exemplifies how our current understanding of protein evolvability can lead to divergent predictions: it could be reasoned that destabilization promoted OmpF$^+$ evolution by increasing the conformational flexibility of the protein, allowing it to occupy unstable, non-native, yet functionally innovative conformers, but it is equally reasonable to predict that because the innovative mutations were destabilizing, they would have been more likely to evolve in a more stable, robust background. We hypothesize that, in the case of λ's receptor use evolution, protein instability was not simply a cost for attaining the OmpF$^+$ mutations, but instead it actually enhanced evolvability by enabling the formation of phenotypically diverse subpopulations of particles with latent new binding abilities.

## Results

### Naturally evolved thermostabilizing mutations reduced host-range expansion evolvability

We initiated our evaluation of the thermostability-evolvability relationship by comparing the propensity to evolve OmpF$^+$ among three OmpF$^-$ precursor strains: one that is unstable and two naturally evolved thermostable derivatives. To generate the unstable precursor strain, we started with a well-studied unstable OmpF$^+$ λ generalist ("7-mut") [26] and engineered out a single mutation (N1107K) that is known to be critical for OmpF function [27]. The resulting "6-mut" genotype was unable to use OmpF and remained as unstable as "7-mut" (Fig 1A), with both less stable than the ancestral λ genotype (Fig 1A). Next, we selected the resulting OmpF$^-$ "6-mut" for increased thermostability without selecting for receptor function (see materials and methods). We found two mutations (T987A and F1122L) that each independently enhanced 6-mut thermostability (Fig 1B), restoring thermostability to that of the ancestral λ (Fig 1B). Intriguingly, the stabilizing mutations bracket the region of J that contained most host-range mutations in the original study documenting the OmpF innovation [25]. Amino acid 987 lies at the very beginning of that "reactive region" but was never mutated in the original study, whereas amino acid 1122 lies at the very end and was mutated in a single isolate from the original study [25]. That mutation, I1122F, was one of the seven mutations in the 7-mut genotype [26] that was used to initiate this study.

To test the effect of stabilization on λ evolvability, we performed evolutionary replay experiments in which the unstable precursor and the two thermostable derivatives were cultured for hundreds of generations. The experimental design (Fig 2A) mirrored the original coevolution experiment in which OmpF function first evolved in the laboratory [25], except that in half of the replicates, we coevolved phage with a host genotype isolated from later in the original experiment, in an effort to correct for the time-shift between phage and bacteria caused by starting with the partially evolved 6-mut. This time-shifted host had a duplication in the *malT* gene, a regulator of LamB expression, causing LamB to only be expressed in rare mutants that had reverted the duplication [28]. All three genotypes were capable of evolving OmpF use, but the two thermostable variants required more time to evolve (6–10 days for the stable genotypes compared to 1–2 days for the unstable genotype), and most replicate populations of the stable

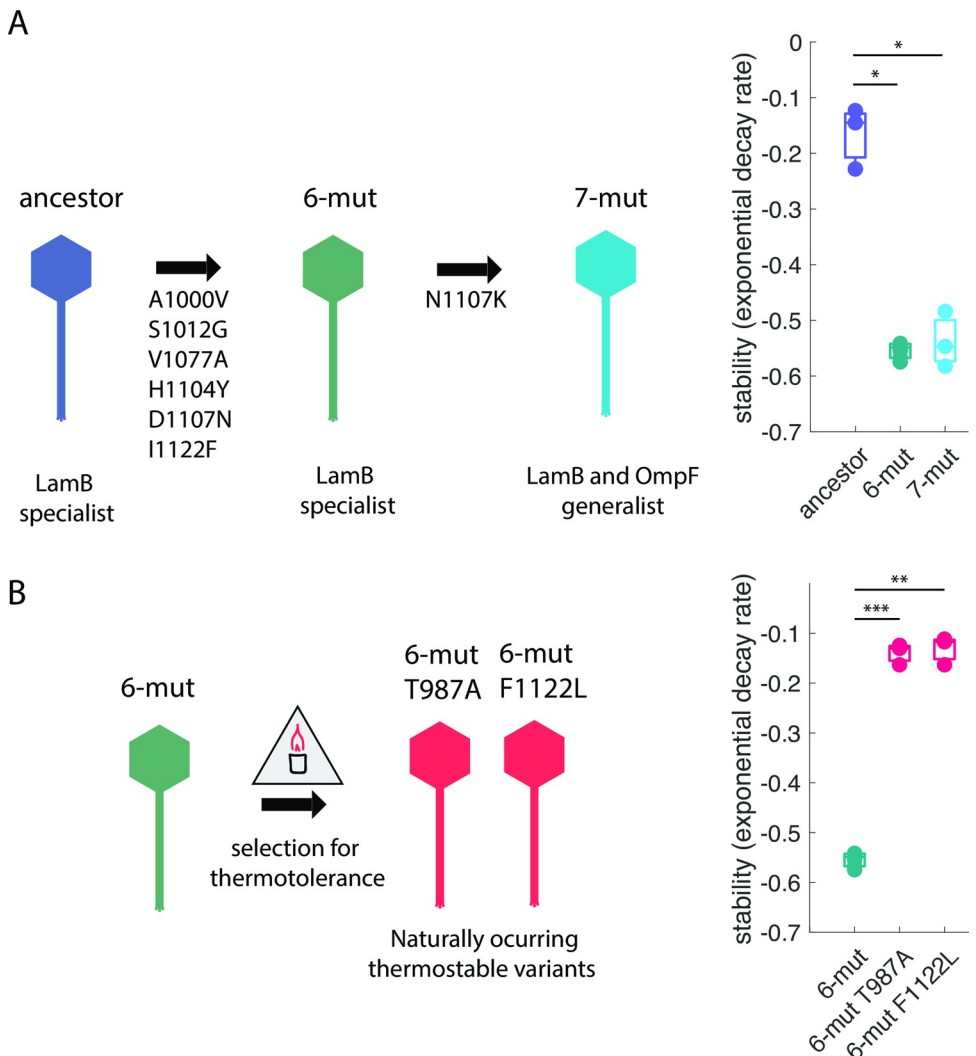

**Fig 1. Experimental system and selection for thermostability. A:** To generate our experimental system, we began with an unstable OmpF$^+$ genotype (7-mut) and edited out a critical mutation (N1107K), creating an OmpF$^-$ genotype that remained unstable and was on the verge of evolving OmpF$^+$. Stability plots: n = 6 replicates for 6-mut, n = 3 replicates for ancestor, and n = 3 replicates for 7-mut. Comparisons of 6-mut and 7-mut to ancestor were made using paired t-tests corrected for multiple comparisons by Bonferroni method (ancestral λ to 6-mut: p = 8.55x10$^{-4}$; ancestor to 7-mut: p = 9.80 x 10$^{-4}$). Bonferroni corrected significance thresholds: ns: p > 0.0024, $^*$: p < 0.0024, $^{**}$: p < 0.00024, $^{***}$: p < 2.4x10$^{-5}$. **B:** We then selected the 6-mut for enhanced thermostability, generating two naturally evolved thermostable genotypes: T987A and F1122L. Stability plots: n = 6 replicates for 6-mut, n = 3 replicates for T987A, and n = 3 replicates for F1122L. Comparisons of 6-mut and 7-mut to ancestor were made using paired t-tests corrected for multiple comparisons by Bonferroni method (6-mut to 6-mut T987A: p = 1.40 x 10$^{-5}$; 6-mut to 6-mut F1122L: p = 2.73 x 10$^{-5}$). Bonferroni corrected significance thresholds: ns: p > 0.0024, $^*$: p < 0.0024, $^{**}$: p < 0.00024, $^{***}$: p < 2.4x10$^{-5}$.

genotypes never evolved during the ten-day experiment (Fig 2B). This result was despite the fact that most replicate populations of stabilized genotypes attained titers equal to or higher than those of the unstable 6-mut (S1 Fig).

Sequencing revealed an explanation for why unstable genotypes evolved OmpF$^+$ faster than thermostable genotypes: a single amino acid substitution allowed the unstable genotype to evolve OmpF$^+$, while thermostable genotypes required two substitutions (Fig 2C). It was not surprising that a single mutation could restore OmpF$^+$ in the unstable 6-mut background, since 6-mut was created by reverting a single mutation from an OmpF$^+$ genotype.

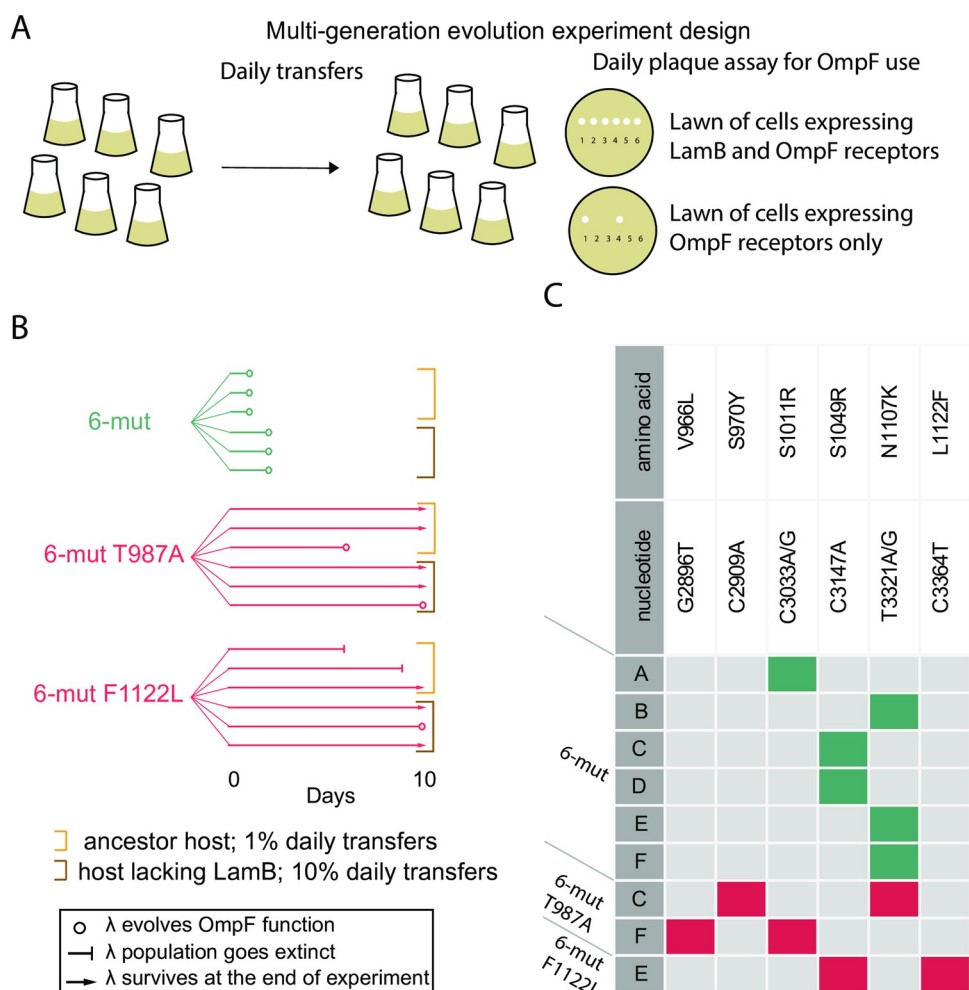

**Fig 2. Evolution experiment with naturally occurring thermostable genotypes. A:** A ten-day evolution experiment was performed to assess OmpF-use evolvability. **B:** Evolutionary trajectories of six replicate populations of each starting genotype are denoted by six parallel lines. When λ evolves the innovation or goes extinct is indicated by symbols indicated in legend. Brackets surround replicates from two different ways of running the coevolution, which did not appear to impact whether OmpF⁺ evolved. **C:** A single plaque from each replicate population was sequenced on the first day OmpF⁺ was detected. Mutations are indicated along the top. Boxes with colored fill indicate that the amino acid change occurred in an isolate. The fill color indicates the stability of the genotypic background in which the mutation evolved (teal = unstable, red = thermostable.) Population IDs are indicated by letters.

Unexpectedly though, only three of the six 6-mut replicates restored the same amino acid that was reverted (1107), while the other three replicates obtained a different mutation (S1011R or S1049R), suggesting the role N1107K plays in OmpF function can be accomplished through other mutations (Fig 2C). Each thermostable genotype that evolved OmpF⁺ also gained either N1107K, S1011R, or S1049R (Fig 2C). Parallel evolution of these four mutations across unstable and thermostable backgrounds suggests that this set of mutations may produce a similar phenotypic effect in J that is required for activity on OmpF, regardless of the thermostability of the genotype in which it occurs. Curiously, another set of mutations (V966L, S970Y, and L1122F) only evolved in thermostable backgrounds and were never detected in the unstable background (Fig 2C). Each instance of OmpF⁺ in a thermostable background required one mutation from each set (Fig 2C).

## Engineering a library of thermostability variants by manipulating a single amino acid

The two naturally selected thermostable λ genotypes provided notable cases in contradiction to the hypothesis that thermostability promotes evolvability and instead pointed to instability as a trait that potentiates functional innovation. However, this pattern was observed in only a small set of variants, and we next sought to test the hypotheses across a larger set of λ variants. To generate additional variants, we focused on codon 987 in J, the site of one of the naturally selected thermostabilizing mutations (T987A). We chose to focus on 987 instead of 1122 because 1122 lies in the C-terminal region in the J protein that is typically associated with receptor binding [29] and OmpF use evolution [25] and was already mutated in the 7-mut, relative to the λ ancestor. We reasoned that further study should focus on variation at amino acid 987 because physical separation from the region previously implicated in host-range evolution might reduce the risk of confounding effects. We used Multiplexed Auotmated Genome Engineering with coselection [30] to create a library of amino acid variants at this position in the 6-mut genotypic background. From this procedure we were able to generate nine of the remaining 18 possible amino acid variants at position 987 (New amino acids L, C, S, G, K, R, Y, I, and P).

We hoped to generate a gradient of stability variants to assess the precise relationship between thermostability and evolvability; however, we were only able to engineer three levels of thermostability: stable, unstable, and catastrophically unstable. Six variants were as stable as ancestral λ (paired t-tests, n = 3 per genotype; T987S: p = 0.697; T987C: p = 0.575; T987Y: p = 0.412; T987R: p = 0.448; T987G: p = 0.503; T987K: p = 0.575, Bonferroni corrected α value = 0.0016) and only one variant was as unstable as 6-mut (T987L: p = 0.510, Fig 3A). One of the remaining variants, T987P, failed to produce any phage upon lysogen induction, and a second variant, T987I, produced a culture with ~$10^{-4}$ of the normal 6-mut titer. Upon sequencing J from plaques that formed from the T987I lysogen, we discovered that all viable descendants had an additional mutation, indicating that the T987I variant can only produce infectious particles if it gains a "rescue" mutation while the phage genome replicates in the lysogenic state. The two rescue mutations identified were A1077V and F1122L. Although we are not able to measure the stability of the T987I variant without the rescue mutations, we can infer that the mechanism of rescue is likely stabilization, since F1122L is stabilizing in the 6-mut genotype background, and A1077V is a reversion of one of the six mutations that the stable λ ancestor evolved *en route* to the unstable 6-mut. We concluded that T987I and T987P might be catastrophically unstable, with J proteins that completely fail to fold, so we excluded these genotypes from further experiments.

## Among the engineered library, most thermostable variants had reduced evolvability

We then measured the evolvability to gain OmpF function for each newly engineered variant in the library. Based on the results from the naturally selected thermostable variants, we predicted that T987L would be the most evolvable of the engineered variants because it had the lowest stability while still remaining viable. To test our prediction, we evolved six replicate populations of each engineered genotype, as well as six replicate populations of the 6-mut as a control, using the same experimental conditions and host strain as the initial experiment. As predicted, T987L was the most evolvable engineered genotype, with all six replicates using OmpF after just one day of evolution (Fig 3B). Among the six thermostable variants the fraction of replicate populations to evolve and number of days required differed widely among variants (Fig 3B). T987G never evolved OmpF⁺, while T987S, T987K, T987R, and T987Y were

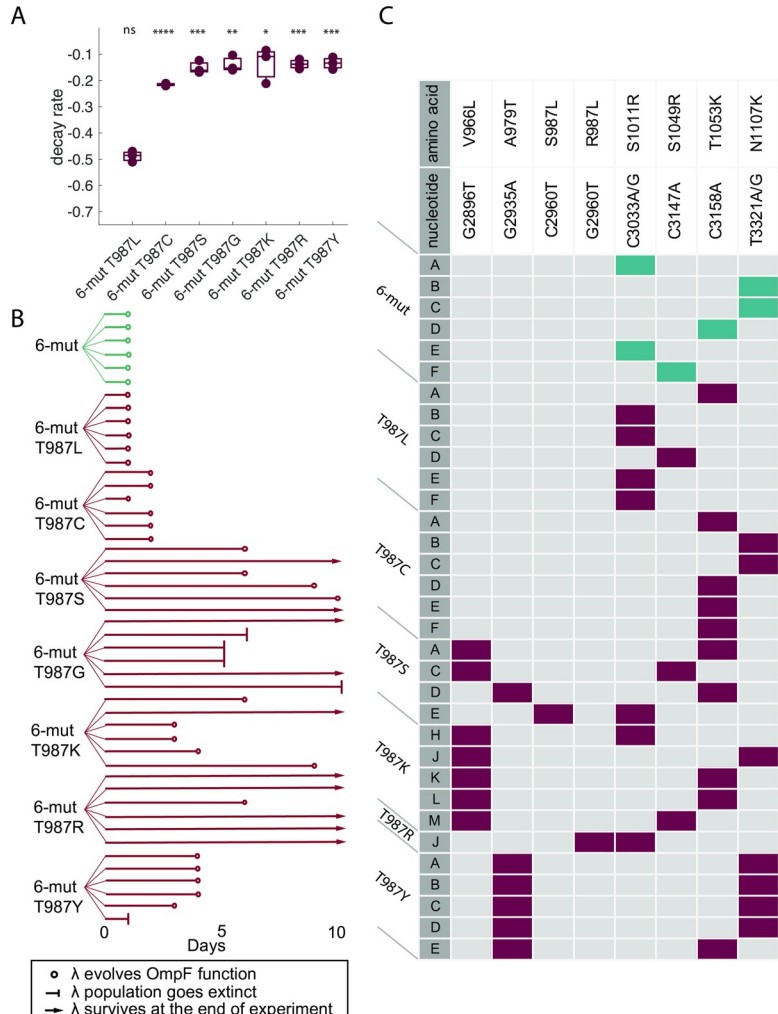

**Fig 3. Evolution experiment with engineered library genotypes.** We generated additional variants of 6-mut differing only at the amino acid at position 987 in J. **A:** Most variants were more thermostable than 6-mut, but T987L remained as unstable as 6-mut. N = 3 replicates per genotype. Comparisons to 6-mut decay rate were made using paired t-tests corrected for multiple comparisons by the Bonferroni method (6-mut T987L: p = 0.510; 6-mut T987C: p = 7.58 x 10⁻⁸, 6-mut T987S: p = 1.02 x 10⁻⁵, 6-mut T987G: p = 1.26 x 10⁻⁴, 6-mut T987K: p = 8.55 x 10⁻⁴, 6-mut T987R: 2.96 x 10⁻⁶, 6-mut T987Y: p = 7.26 x 10⁻⁶.) **B:** We measured the evolvability of OmpF⁺ in each variant using a nearly identical evolution experiment as in Fig 2A and 2B. **C:** A single plaque from each replicate population was sequenced on the first day OmpF⁺ was detected. Mutations are indicated along the top. Boxes with colored fill indicate that the amino acid change occurred in an isolate. The fill color indicates the stability of the genotypic background in which the mutation evolved (teal = 6-mut, dark red = engineered codon 987 variants.) Population IDs are indicated by letters. Asterisks along the lines indicate significant differences in decay rate between the genotypes connected by the line (Bonferroni adjusted significance: ns: P > 0.05, *: P <0.05, **: P < 0.01, ***: P< 0.001, ****: P < 0.0001).

able to evolve OmpF⁺ but fewer replicate populations evolved and required longer than the 6-mut and T987L (Fig 3B). A notable exception to the overall pattern was T987C, which was the only thermostable variant with high evolvability, requiring just one or two days of evolution (Fig 3B).

As with the initial experiment, we sequenced a single clone from each isolate that evolved OmpF⁺ on the first day that plaques were visible on the LamB knockout lawn. And consistent with the initial experiment, the time to evolve OmpF⁺ was explained by the number of mutations required. Each of the 6-mut control populations required a single mutation for OmpF⁺,

from the same set of three mutations (S1011R, S1049R, and N1107K) as the initial experiment, except that one population evolved OmpF+ by a single mutation (T1053K) that was not previously detected as a N1107K substitute (Fig 3C). And T987L, the only unstable engineered variant, also evolved OmpF+ in all six replicates with a single mutation, either N1107K or one of the same substitutes (Fig 3C). Among the stable engineered variants, T987S, T987K, T987R, and T987Y required two mutations for OmpF+, while the variant that evolved more quickly (T987C) required only one (Fig 3C). We will return to potential explanations for this outlier in a subsequent section, but for now we will discuss the structural and functional roles of the two sets of mutations.

## J structural prediction provides insight into mutation function

We wondered whether the pattern of mutation observed in the coevolution experiments might shed light on a structural mechanism behind J evolution. Perhaps evolution to use OmpF involves manipulating both the conformational flexibility of J, in addition to altering the very specific residues that make direct contact with OmpF. Recall that all variants, regardless of thermostability, require either N1107K or a substitute mutation, whereas thermostable variants required an additional mutation, typically toward the N-terminus of the protein. We predicted that N1107K and substitutes are the OmpF contacting residues, and presumably would lie clustered on one surface of J. Under this model, the additional mutations, required only by the thermostable variants, destabilize J, and thus increase conformational flexibility. We predicted that these mutations would lie embedded within the protein. Because the J protein structure remains unsolved experimentally, we used AlphaFold, a new technology that uses machine learning to predict protein structures with remarkable accuracy [31], to predict the structure of the reactive region of the 6-mut genotype (predictions for the ancestor and 6-mut were nearly identical, S2 Fig). We modelled the protein multiple times by varying which segments of the protein to include. In the end, we found that the segment with the highest confidence score and for which all the mutations being studied were included, was a 173 amino acid segment at the C-terminal end. The problem with predicting a larger portion of the protein is that from ~amino acid 780 to 960, the model confidence is very low (40–60%) compared to the terminal region (80–90%). The J region we chose is known through biochemical assays to be the portion that binds the receptor [29], it encompasses the majority of host-range altering mutations previously reported [25], and has been shown to have an elevated rate of evolution in nature [27]. As we expected, N1107K and its substitutes all lie in loops at one surface of the protein, while the thermostabilizing mutations and putative destabilizing mutations all lie at the other end of a series of beta sheets (Fig 4). It is notable that this spatial separation is not simply a reflection of proximity in the peptide sequence but are likely clustered because of 3D position-function correlations. For example, S1011R and N1107K are proximate in the folded structure despite being 96 residues apart, and the same is true for T987A and F1122L (135 residues apart).

## Reconstructing J molecular evolution

To experimentally test this two-step model of protein functional evolution, we reconstructed the evolutionary sequence that started from four different genetic backgrounds and ended with an OmpF+ λ (Fig 2C: pop. B initiated from 6-mut, pop. C initiated from 6-mut T987A, pop. E initiated from 6-mut F1122L; and Fig 3C: pop. E initiated from 6-mut T987S) and measured how stability changed. We included the wild type pathway from 6-mut to 7-mut via N1107K for comparison, as we did not expect this pathway to be associated with a change in stability, and indeed there was no effect (Fig 5). In line with our prediction, the putative

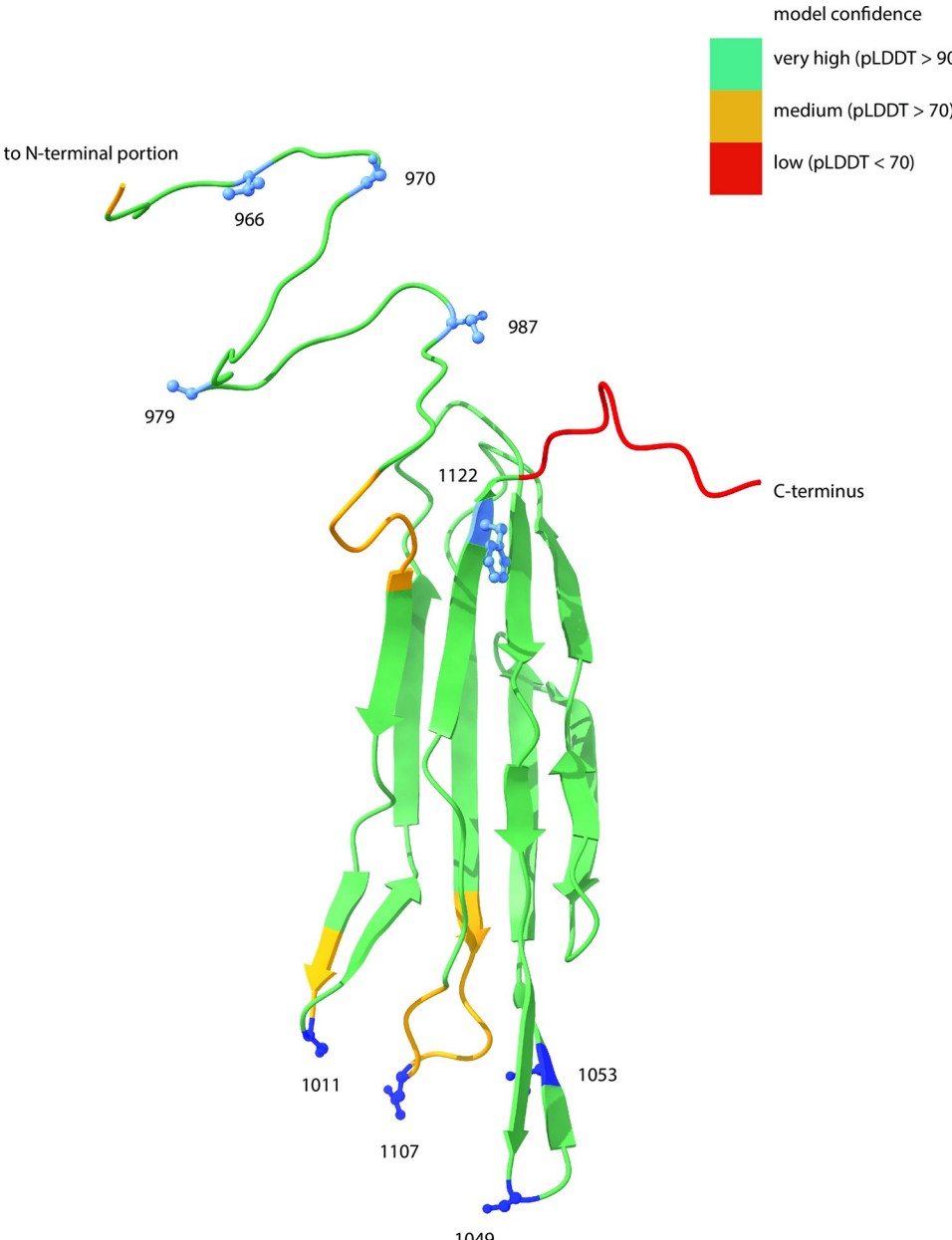

**Fig 4. Mapping thermostabilizing and putative destabilizing mutations on a structural prediction of J's reactive region.** We used Alphafold [31] to predict the structure of the reactive region of J (amino acids 960–1132). This structure corresponds to the 6-mut genotype, but the 6-mut prediction was nearly identical to the ancestor prediction (S2 Fig). We then mapped the surface binding mutations (dark blue residues) and the thermostabilizing and destabilizing mutations (light blue residues) onto the structure. Coloration of the backbone indicates model confidence.

destabilizing mutations (S987L, L1122F, and S970Y) were in fact destabilizing in the background in which they occurred, although the destabilizing effect was stronger in the 6-mut F1122L and 6-mut T987S backgrounds compared to the 6-mut T987A background (Fig 5). Notably, the destabilizing mutation that occurred in the 6-mut F1122L background was an exact reversion of the stabilizing mutation (L1122F, Fig 5). As expected, the addition of the

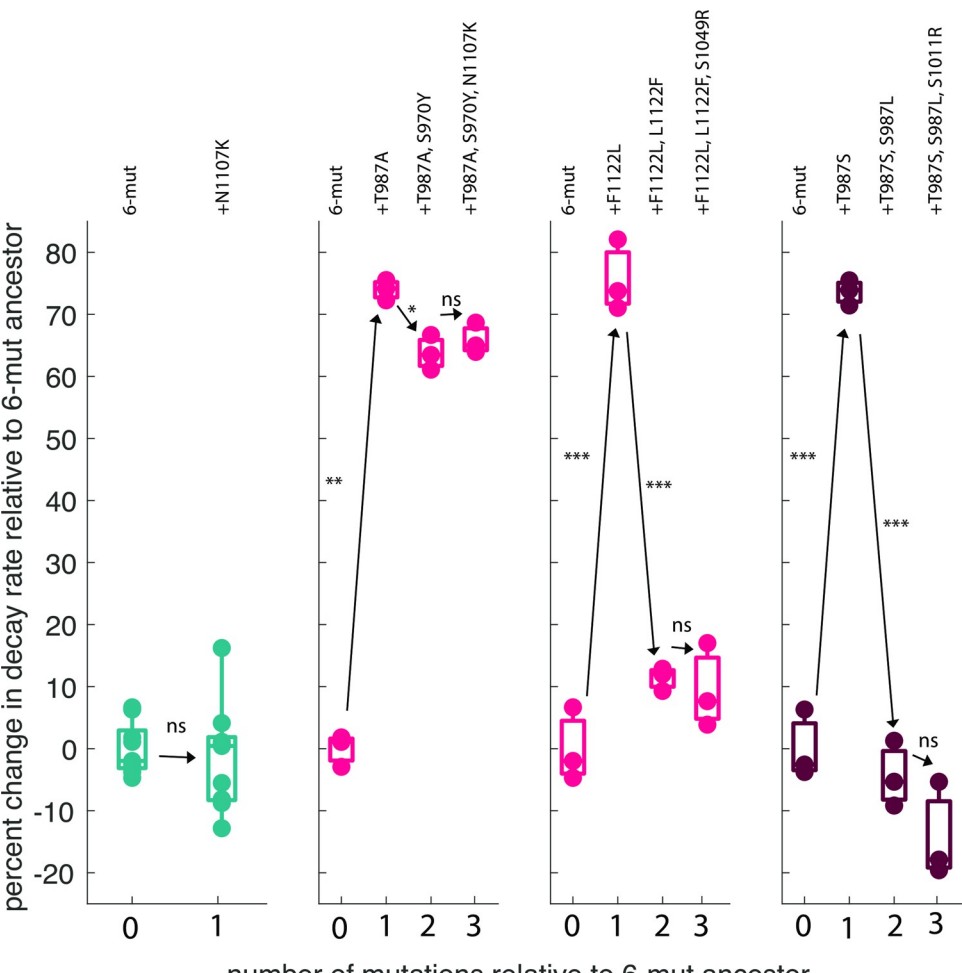

**Fig 5. Trajectories of evolution of OmpF$^+$ in unmodified and stabilized backgrounds.** Evolutionary trajectories of selected isolates from the replay experiment, reconstructed with genomic engineering. In the 6-mut (teal), a single mutation led to an OmpF$^+$ genotype. In three stabilized backgrounds (naturally evolved = red, engineered = dark red), an additional destabilizing mutation was required as a steppingstone to the OmpF$^+$ genotype. Asterisks along the lines indicate significant differences in decay rate between the genotypes connected by the line, as compared by paired t-tests corrected for multiple comparisons using the Bonferroni method. First panel: 6-mut vs. 6-mut N1107K: p = 0.678; Second panel: 6-mut vs. 6-mut T987A: p = 1.88x10$^{-6}$; 6 -mut T987A vs. 6-mut T987A S970Y: p = 0.0055; 6-mut T987A S970Y vs. 6-mut T987A S970Y N1107K: p = 0.3818; Third panel: 6-mut vs 6-mut F1122L: p = 9.25x10$^{-5}$; 6-mut F1122L vs. 6-mut F1122L L122F: p = 5.13x10$^{-5}$; 6-mut F1122L L122F vs. 6-mut F1122L L1122F S1049R: p = 0.667; Fourth panel: 6-mut vs. 6-mut T987S: p = 2.65x10$^{-5}$; 6-mut T987S vs. 6-mut T987S T987L: p = 1.83x10$^{-5}$; 6-mut T987S T987L vs. 6-mut T987S T987L S1011R: p = 0.143. (Bonferroni adjusted significance: *: p < 0.0167, **: p < 0.00167, ***: p < 0.000167, ****: p < 1.67e-5).

putative contact surface mutation did not alter viral particle stability in any reconstructed background (Fig 5).

## A novel molecular mechanism for host-range evolvability

We propose a multiple phase model of J evolution consistent with the sum of our findings. In the first phase, ancestral J gains destabilizing mutations that alter its folding dynamics such that a portion of the expressed proteins fold into an alternative, non-native conformer (Fig 1A, ancestor ➜ 6-mut). The next phase of mutations alters the binding surface to facilitate specific surface amino acid interactions between J and OmpF (Fig 1A, 6-mut ➜ 7-mut). The catch

with this model is that the second phase mutation only works if it occurs in an unstable protein that expresses some fraction of a non-native conformer that places key amino acid residues in proximity to binding partners on the OmpF molecule. If the stabilizing mutations increase protein rigidity and suppress the formation of non-native conformers, then even if the N1107K mutation occurs, the residue would not be in the correct position to interact with its partner on OmpF. To test this, we engineered the surface mutation N1107K into all variants and tested for OmpF activity. As expected, both unstable backgrounds and the outlier T987C became OmpF$^+$, whereas none of the remaining stable backgrounds yielded strong OmpF use (S1 Text, S3 Fig, S1 Table). Counter to our expectations, though, editing in N1107K into two of the stable backgrounds, T987A and T987Y, yielded very weak activity on OmpF. The edited genotypes were unable to produce plaques on a lawn of *lamB*⁻ cells; however, they produced faint clearing at high phage concentrations spotted on *lamB*⁻ lawns (S1 Text and S3 Fig). We verified that faint clearings on plates corresponded to weak growth on *lamB*⁻ cells in liquid culture (S1 Text and S4 Fig). This result suggests that the steps in the molecular model are not as discrete as anticipated, yet the path to strong OmpF use does involve both destabilizing mutations and binding surface alterations.

## T987C: The thermostable, yet evolvable variant

While our model was able to correctly predict the evolvability of six engineered strains, it failed to predict the high evolvability of T987C, the only variant to be both highly thermostable and highly evolvable. To first verify that this outlier did not result from additional mutations outside *J* that might have occurred during library creation and affected OmpF$^+$ evolution, we generated an independently engineered T987C variant and re-measured it's evolvability with 12 replicate populations, along with 12 replicate populations of 6-mut as a control. All 12 populations of T987C evolved OmpF$^+$ on the first day (S5 Fig), consistent with the initial experiment, even slightly faster, ruling out this possibility. At present, our best hypothesis to reconcile T987C with our model is that there may be rare genotypes that are thermostable but produce sufficient copies of the non-native, OmpF-binding J to complete infection, bypassing an extra mutation that yields non-native conformers via destabilization. If a destabilized J protein is not strictly required for evolving OmpF-binding, as suggested by the T987C variant, this raises the question of whether there may be mutational pathways available to the ancestral $\lambda$ genotype that do not include destabilizing mutations before receiving the surface contact mutation. Such pathways, if they exist, should be highly favored because intermediate genotypes would not pay the cost of low thermostability. One explanation for why $\lambda$ might have originally evolved via destabilization despite the cost of instability is that the simultaneous optimization of evolvability and thermostability instigates a trade-off with other traits that determine fitness. Previous studies on phages have shown fitness tradeoffs between thermostability and growth rate [32,33] that could close off pathways that include evolving stabilizing mutations in competitive environments, making variants like 6-mut T987C accessible via engineering, but not through adaptive evolution. To test this idea, we measured the growth rates of all variants on the same host as was used in the evolution experiment, and T987C had the lowest average growth rate of any of the variants, significantly lower than that of its 6-mut progenitor (Fig 6). This result suggests that the two-phase model of molecular innovation outlined above emerged from a combination of biophysical and adaptive constraints.

## Evaluating alternative models of protein evolution

Our hypothesis for why some variants required two mutations to evolve OmpF$^+$ is that they required an extra destabilizing mutation to restore evolvability. An alternative possibility is

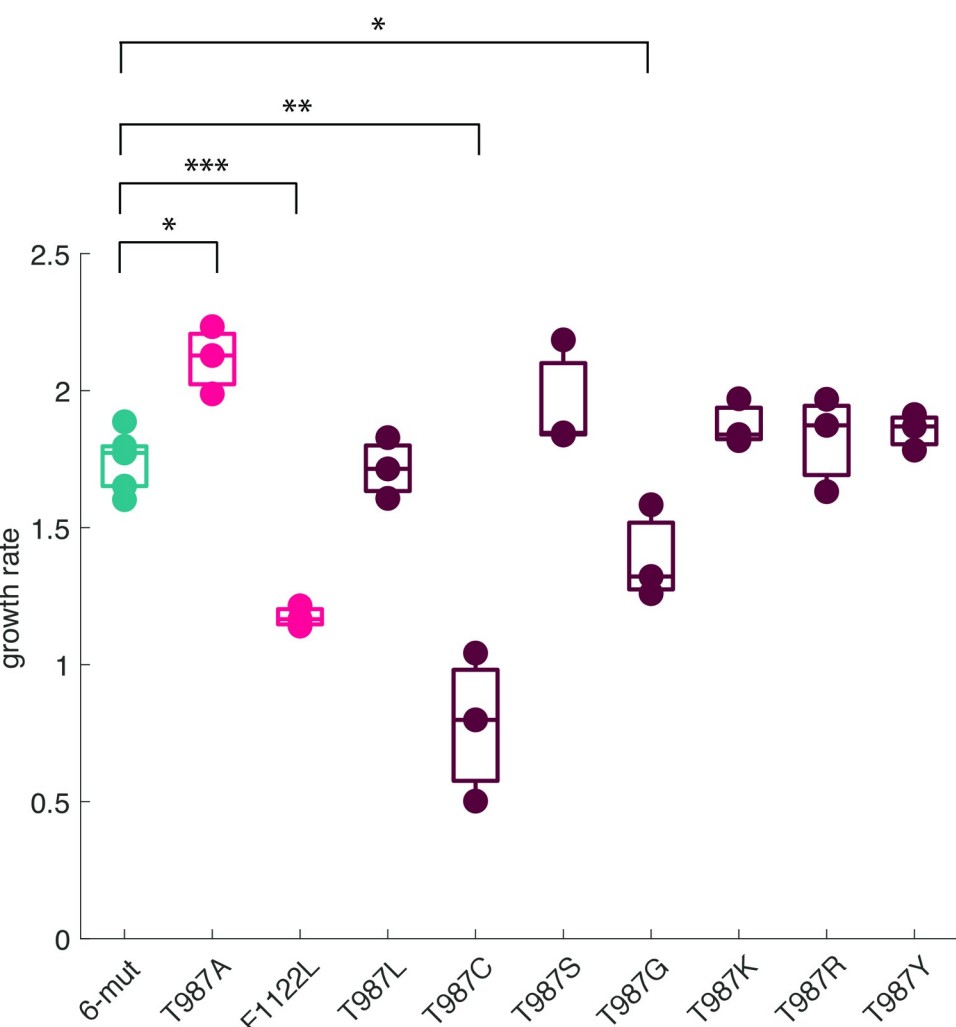

**Fig 6. Growth rates of naturally occurring thermostable genotypes and engineered library genotypes.** Growth rates on REL606 in M9 glucose + $MgSO_4$ media over four hours. Comparisons to 6-mut growth rate were made using paired t-tests corrected for multiple comparisons using the Bonferroni method, N = 3 per genotype. Significantly higher: T987A: p = 0.002; significantly lower: F1122L: p = $4.12 \times 10^{-5}$; T987G: p = 0.0052; no difference: T987L: p = 0.695; T987S: p = 0.068; T987K: p = 0.105; T987R: p = 0.419; T987Y: tstat = p = 0.150 Bonferroni adjusted significance: ns: P > 0.0056, *: P < 0.0056, **: P < 0.00056, ***: P < 0.000056.

that the destabilizing effect is coincidental, and the true function of the extra mutation is to compensate for fitness costs incurred by the stabilizing mutations in the 6-mut background. Recall that this is unlikely since most of the less evolvable stabilized variants had growth rates greater than or equal to that of 6-mut (Fig 6).

Along similar lines of thinking, perhaps the extra mutation is required to ameliorate a genetic incompatibility between the stabilizing mutations and the mutations found in the contact surface. This could occur if stabilizing mutations and surface mutations are beneficial alone but produce a non-functional protein when they co-occur in the same genotype. To rule out this possibility, we measured productivity of lysogens for the naturally selected thermostable variants (T987A, F1122L) with and without N1107K and found no difference in titer (S6 Fig) indicating that there is not a genetic incompatibility [34].

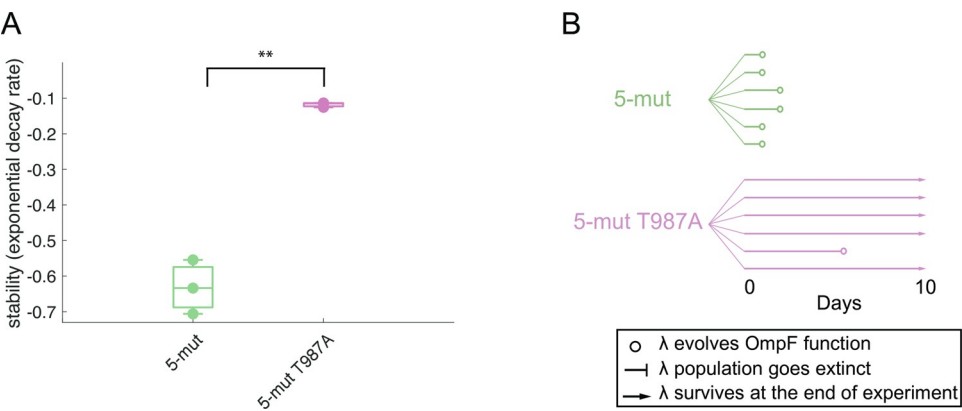

**Fig 7. Stability and evolvability of 5-mut and 5-mut T987A stable variant. A:** In a background that is further away from evolving OmpF+ (two mutations reverted compared to one mutation in 6-mut), the T987A mutation is still stabilizing. **B**: In the 5-mut background, T987A reduces evolvability to an even greater extent. Two sample t-test for unequal variance, N = 3 per genotype, **: P = 0.0069.

Another possibility is that the stabilizing mutations interfere specifically with the very last step in OmpF$^+$ evolution (the N1107K mutation), but had they been introduced in a background further removed from OmpF$^+$, they would have enhanced evolution. To test this hypothesis, we generated a new background, a 5-mut created by reverting another 7-mut mutation by changing the amino acid at codon 1012 from G back to S. We then edited in the T987A stabilizing mutation and confirmed that it does stabilize the 5-mut as expected (Fig 7A). To measure evolvability, we ran a 10-day coevolution experiment and found that all six populations of the unmodified 5-mut genotype evolved in the first two days of the experiment, while only one out of six populations of the stabilized 5-mut evolved OmpF$^+$, requiring 5 days (Fig 7B). Under the conventional model of protein stability and evolvability, stabilizing the J protein should have increased the J protein's capacity to search for adaptive mutations. This effect should have been enhanced when starting further from the innovation; however, we found the opposite. The difference in evolvability between the unmodified 5-mut and the stabilized 5-mut was equal, if not slightly greater, than the difference in evolvability between the unmodified 6-mut and the stabilized 6-mut.

## Discussion

We found that, on average, increasing the thermostability of λ's host-recognition protein resulted in the loss of evolvability, which was restored when thermostable genotypes evolved destabilizing mutations. This pattern is the opposite of the prediction that could be made based on the preponderance of current literature—that inserting thermostabilizing mutations should enhance evolvability. This current consensus is rooted in an observed trade-off between thermostability and function in proteins: mutations that confer new functions often destabilize the native fold as a side effect [10,35]. The stability-function trade-off has presented a particular challenge for directed evolution, which requires multiple rounds of mutagenesis and selection [36]. One solution that is thought to be relatively drawback-free has been to select for naturally occurring stabilizing mutations or deliberately engineer known stabilizing mutations into enzymes either before or between cycles of directed evolution [7,36,37]. The robustness conferred by extra thermostability preserves the native fold and function while allowing for new functions to be explored.

In the case of J, mutations that led to gain-of-function on the OmpF receptor were destabilizing, indicating that the J protein is also subject to the stability-function trade-off. However,

inserting thermostabilizing mutations into an unstable J genotype on its way to evolving expanded host-range actually slowed or completely stalled evolution. This indicates that J discovered an alternative solution to the stability-function trade-off. Based on our results and prior work [26], we suggest that unstable genotypes circumvent the trade-off by producing multiple conformers from a single peptide sequence, at least one of which retains preferential binding to the ancestral LamB receptor [26]. If a genotype produces a high fraction of thermostable, LamB-binding particles, the less stable, non-ground state conformers would be released from the selective pressure to maintain LamB function and free to optimize latent binding interactions with new partners. We propose that it is specifically the lack of structural rigidity in the 6-mut genotype and variant T987L that enhanced the evolvability of those genotypes.

Why might some proteins solve the stability-function trade-off with robustness and others with conformational flexibility, as in the case of λ? The lack of a universal pattern suggests that the relationship between thermostability and evolvability is multifaceted. We can only speculate on the true nature of this relationship, but our leading idea is that each protein exists on a spectrum of thermostability, and extremes at either end can inhibit evolvability. Indeed, we found that one variant from our engineered library, T987I, was likely too unstable to fold, requiring a rescue thermostabilizing mutation to produce infectious particles. Unfortunately, none of the variants fell between T987I and T987L in thermostability, so we cannot precisely identify the inflection point where instability is so severe that evolvability is inhibited. Similarly, it is also unclear at what point increasing thermostability might begin to inhibit evolvability. Additionally, prior work on this question has focused primarily on enzyme proteins, and it is not entirely surprising that the paradigm that emerged from those studies might not apply to viral attachment proteins, especially if viral proteins generally have different properties from cellular proteins [38]. Given these unknowns, it is not surprising that different studies have reported both positive and negative relationships between thermostability and evolvability. Even within a single protein, we did not find a universal pattern, as evidenced by the clear exception of the T987C variant. The balance between robustness and conformational flexibility may also depend upon the extent of structural change required to achieve novel functionality (i.e. localized amino acid substitutions vs. conformation changes). It is possible that for some proteins, single amino acid substitutions that subtly alter active sites or binding surfaces may be sufficient to confer new functions, whereas in other proteins, more global shape changes are required. We suspect that J's evolution from OmpF$^-$ to OmpF$^+$ falls into the latter category, and single mutations such as N1107K, do not confer OmpF$^+$ unless they occur in a background with conformational flexibility. Under this model, for most of the thermostable variants, the increased structural rigidity caused by stabilizing mutations [39] would stifle evolvability by locking J into a single conformation. This interpretation requires the caveat that we did not perform structural studies on J, but the predicted structure of J's reactive region is consistent with our model.

While thermostability explained much of the variation in evolvability, it was not fully predictive. Two mutations, T987L and T987C, differed greatly in thermostability (Fig 3A), yet had nearly identical evolvability, both evolving OmpF$^+$ in all populations in 1–2 days with one mutation (Fig 3B and 3C). Similarly, among a cluster of engineered variants that had indistinguishable thermostabilities (T987S, T987G, T987K, T987R, T987Y, Fig 3A), we observed a breadth of responses in evolvability, as measured by the number of days required to evolve OmpF$^+$, the fraction of populations that eventually evolved, and the number of unique 2-mutation pathways taken during evolution (Fig 3B and 3C). There are two insights that can be drawn from these observations. First, the ability to evolve OmpF$^+$ did not hinge on the specific residue at position 987; instead, two residues besides the wild type could be interchanged without impacting evolvability. And second, viruses differing only at a single residue in a single

protein were surprisingly variable in their capacity for host-range expansion, and differences in key traits like thermostability and growth rate were only partially predictive. That our library variants were genetically identical except for the amino acid at a single site was both a strength and a limitation of our study. This design allowed us to pinpoint and precisely manipulate the genetic basis for stability and host-range evolvability in λ; however, focusing in on a single site might result in non-independent effects, and conclusions drawn from a single protein might reflect idiosyncrasies of its particular evolutionary history.

In this study, we evolved engineered viruses under highly controlled laboratory conditions to examine how thermostability affects host-range evolvability, allowing us to precisely manipulate our variable of interest. Such a design cannot, and was not intended to, recapitulate the complexity of nature. However, prior work indicates that the evolution of *λ*'s J gene in the lab may be uniquely informative for understanding evolution in nature due to a remarkable parallelism between sites under selection in the laboratory and in natural populations [27]. The entire reactive region of J, encompassing the vast majority of mutations that evolve in during OmpF gain-of-function experiments, was shown to be a hotspot for molecular evolution in natural J homologs, and amino acids 1012 and 1107 had particularly high rates of evolution [27]. Whether this parallelism between laboratory and nature extends to stabilizing mutations is unknown, but our simple selection identified mutations that affected stability at a physiologically relevant temperature (37°C), and it is plausible that the same mutations might aid virus survival under natural conditions. That said, our results are based upon a relatively small number of genotypes in a single experimental system, and additional work will be required to assess the generality of the pattern and its use in predicting host-range expansions.

Understanding the predictors of evolvability has important applications, such as providing insights into the characteristics that potentiate viral host shifts. There has been considerable interest in conducting surveillance on viral populations with the goal of detecting variants with emergence potential before they first spill over into human populations [40]. This type of surveillance would be aided by information about the characteristics of viruses that influence the likelihood of host-range evolution. Our results were derived from a study of bacteria-infecting virus rather than an animal virus, and further work is warranted to investigate the role of viral particle instability in animal virus host-range evolvability. Viral instability can be rapidly assessed in the laboratory, or if only genome sequences are available for given viruses, there are bioinformatic tools capable of predicting protein stability [41]. This work was completed during the 2020–2021 coronavirus pandemic, raising the question of whether instability in SARS-CoV-2 host recognition protein (S) aided its transition to humans. At the time of publication this had not been tested; however, a mutation that evolved during the pandemic stabilizes the SARS-CoV-2 S protein [42] and also increases viral titer [43], suggesting that the variant that jumped to humans was relatively unstable.

## Materials and methods

### Media

We used the following recipes to prepare media: **LB Lennox broth**: 20 g LB Lennox powder per liter of water. **LBM9**: 20 g tryptone, 10 g yeast extract per liter of water supplemented with 47.7 mM disodium phosphate heptahydrate, 22.0 mM potassium phosphate monobasic, 18.7 mM ammonium chloride, 8.6 mM sodium chloride, 0.2 mM calcium chloride and 10 mM magnesium sulfate. **M9 glucose:** 47.7 mM disodium phosphate heptahydrate, 22.0 mM potassium phosphate monobasic, 18.7 mM ammonium chloride, 8.6 mM sodium chloride, 0.2 mM calcium chloride, 10 mM magnesium sulfate, 5.55 mM glucose, 7.54 μM thiamine, and 0.02% LB. **M9 minimal glycerol:** 47.7 mM disodium phosphate heptahydrate, 22.0 mM potassium

phosphate monobasic, 18.7 mM ammonium chloride, 8.6 mM sodium chloride, 0.2 mM calcium chloride, 10 mM magnesium sulfate, 7.54 uM thiamine, 17.1 mM glycerol, 1.22 mM deoxygalactose, and 0.02% LB. **LB agar:** 10 g tryptone, 5 g yeast extract, 5 g sodium chloride 16 g agar per liter of water. **MacConkey agar galactose:** 40 g MacConkey base and 10 g galactose per 1 liter of water. **Soft agar:** 10 g tryptone, 1 g yeast extract, 8 g sodium chloride 7 g agar per 1 liter of water, supplemented with a final concentration of 2 mM calcium chloride 5.55 mM glucose, and 10 mM magnesium sulfate.

### $\lambda$ Phage strains

The phage used in this study were all derived from the 7-mut $\lambda$ strain described in [26]. The 7-mut is a derivative of the cI857 $\lambda$ lysogen integrated into the HWEC106 genome. HWEC106 has two features that allow genome editing: a deletion in the *mutS* mismatch repair gene and the pKD46 $\lambda$ red recombineering plasmid [44]. Plasmids were maintained by addition of 50 $\mu$g/mL ampicillin to lysogen cultures. Phage particles were produced from lysogen stocks as described in the section "induction of lysogens by heat shock."

### Host *E. coli* strains

For the coevolution replay, we used REL606 and its *malT⁻* derivative, LR01 [28], which was isolated from a previous coevolution experiment [25] and has a 25-base duplication in *malT*, which results in the loss of LamB expression except in mutants that revert the duplication. For density measurements during the replay, we used the Keio knockout collection parental wild type strain BW25113 [45]. For detection of OmpF⁺ mutants during the replay, we used the *lamB⁻* strain (JW3996) from the Keio collection [45]. For all other experiments, we used Keio knockout collection strains (wild type BW25113 and *lamB⁻* JW3996) for all culturing and plating. Throughout this manuscript we refer to BW25113 as "WT".

### Sanger sequencing

We sequenced the region of the *J* gene known to interact with the LamB receptor (approximately position 2650 to 3399 of 3399 total bp) to identify stabilizing mutations, confirm genetic edits, and identify mutations in genotypes that evolved to be OmpF⁺ in the replay experiment. We submitted unpurified PCR products (primers: Forward 5' CCT GCG GGC GGT TTT GTC ATT TA; Reverse 5' CGC ATC GTT CAC CTC TCA CT) to Genewiz La Jolla, CA, for sequencing with the reverse primer. We aligned sequences using Unipro UGENE v1.31.1 [46].

### MAGE to produce 6-mut lysogen

We used MAGE (Multiplexed Automated Genome Engineering) [47,48] to edit a single reversion into the J gene of the 7-mut OmpF⁺ lysogen. We reversed the mutation at position 3321 from A back to the ancestral nucleotide, T. We call this new genotype '6-mut'.

### Induction of lysogens by heat shock

The night before induction, we grew lysogens in LB at 30˚C shaking overnight. The next morning, we inoculated 140 $\mu$L overnight culture into 4 mL LBM9 supplemented with 40 $\mu$L MgSO$_4$, grew at 30˚C shaking for 2 hours, heat shocked at 42˚C for 15 minutes, then incubated at 37˚C shaking until lysates became clear (90 minutes).

## Selection for thermostabilizing mutations

We generated 6-mut phage by inducing the 6-mut lysogen, as described in the section 'Induction of lysogens by heat shock". The lysate was filtered through a 0.22 $\mu M$ syringe filter, diluted in 0.8% NaCl, and plated with 100 $\mu$L WT cells infused in 4 mL soft agar. We then picked 6 plaques from the overlay plate and incubated them with 100 $\mu$L WT in 4 mL LBM9 and 40 $\mu$L MgSO$_4$ at 37˚C shaking overnight. The next morning, plaque cultures were centrifuged and filtered through 0.22 $\mu M$ syringe filters to remove cells. Phage were then incubated at 55˚C, a temperature much higher than their normal growth temperature of 37˚C, for 6 hours. After incubation, 3.3 mL from each phage stock were plated with 250 mL WT infused in 10 mL soft agar. The phage that were still able to form plaques after the 55˚C incubation were selected as candidate thermostable mutants and were further screened for enhanced stability as follows. Plaques were picked into 0.8% NaCl and incubated with 100 $\mu$L WT cells in 4 mL LBM9 and 40 $\mu$L MgSO$_4$ at 37˚C shaking overnight. The next morning, we added 100 $\mu$L chloroform to each culture and centrifuged at 357× g for 20 min. As a secondary screen to verify the increased heat tolerance of the putative thermostable mutants, we incubated phage 58˚C for 1 hour and plated densities before and after incubation. Out of four candidates, one exhibited increased thermostability in the secondary screen and was isolated for sequencing. We identified a single base pair substitution at position 2959 (codon 987) of *J* resulting in an amino acid change from threonine to alanine.

We identified a second stabilizing mutation using a very similar protocol with minor procedural variations. Before heat selection, plaques were grown up for 7 hours instead of overnight, and after heat selection, 4 mL were plated instead of 3.3 mL. Survivors of heat selection were picked into NaCl and re-plated on WT lawns, then a plaque was picked and grown up for 6 hours for the secondary stability screen (whereas for the isolation of the first stabilizing mutation, the plaques were directly grown up instead of being re-plated first). The secondary screen for stability was conducted at 56˚C for 1 hour (instead of 58˚C for 1 hour). Using this slightly modified protocol we identified the stabilizing mutation at position 3364 in *J* (codon 1122) changing the amino acid from phenylalanine to leucine.

## CoS-MAGE to insert stabilizing mutations into 6-mut lysogen

To rule out the possibility that mutations outside J could be driving the increased stability, we edited each stabilizing mutation into a lysogenic strain of 6-mut λ that had not undergone the selection for increased thermostability. We designed oligos for each stabilizing mutation and used CoS-MAGE [30], a genomic editing tool, to insert each mutation into the 6-mut lysogen. To increase the likelihood of picking successful colonies, we co-transformed with an oligo containing a nonsense mutation in the *galK* gene (*galK⁻* oligo, S2 Table). Successful recombination with this oligo resulted in a color change when cells were plated on MacConkey agar plates containing galactose as the only sugar source. The oligos for stabilizing mutations and the *galK* oligos were both used at a final concentration of 5 $\mu$M.

We also attempted to edit in a control mutation in codon 987 that we predicted would be neutral with respect to stability. We intended the control mutation to be conservative with respect to the chemistry of the amino acid side chain, so we chose threonine to serine (T987S) because both have polar, uncharged side chains. However, we found that this mutation unexpectedly had a stabilizing effect, similar to the T987A mutation. We proceeded with all three mutations; two uncovered through selection and the third through intentional engineering.

## CoS-MAGE to create a variant library at codon 987

To generate additional $\lambda$'s varying at the amino acid at position 987 in the J protein, we used an oligo pool containing random amino acids at nucleotide positions 2959, 2960, 2961, the

codon for amino acid 987. We then carried out CoS-MAGE [30] with our oligo pool plus the *galK⁻* oligo (Table S2) by the same protocol as described in the previous section, using the 6-mut lysogen as the recipient strain. To enrich the population for successful recombinants, we selected for conversion of the galK state by growing up the recovered populations in M9 minimal glycerol media containing deoxygalacose as the selective agent. After enriching the population, we plated cultures on MacConkey agar plates containing galactose and picked white colonies (nonfunctional galK) for sequencing. Approximately one third of the colonies had mutations at codon 987, and we grew these up overnight to create lysogen stocks. All library strains were stored in the lysogenic state at -80˚C, and lysogens were induced to create fresh phage stocks for all further experiments.

### Decay assays

Decay assays were performed identically for all genotypes, and three replicates were included for each genotype. On the day of the assay, phage were induced from lysogens as described in the section 'Induction of lysogens by heat shock' with three replicates per genotype. Lysates were then filtered through 0.22 $\mu M$ syringe filters. Filtered lysates were immediately serially diluted in LBM9 and incubated in glass culture tubes in a warm water bath at 37˚C for 6 hours, quantifying surviving phage at t = 0 (just before starting the incubation) and t = 6 hours. Measurements were taken from repeated sampling of the same tube. Quantification was done by overlay plating 4 mL molten soft agar with 100 uL of WT bacteria + 10 or 100 uL phage lysate over an LB plate.

To measure stability, we calculated overall exponential decay rates from t = 0 to t = 6 hours ($d_{06}$). To compare decay rates among genotypes, we used one-way ANOVAs followed by post-hoc Tukey HSD test (n = 3 for each genotype).

We were not able to include all genotypes in a single experiment on the same day, so we carried out experiments in a blocked design such that each set of measurements included 6-mut replicates, as a way of gauging how much measurements varied across days. For the majority of blocks, no day effects were detected; however, there were a few cases where the measurements for 6-mut stability slightly deviated. To ensure that day-effects were not driving significance in comparisons between genotypes, statistical comparisons of each genotype to 6-mut were made only between replicates measured on the same day (paired t-tests corrected by the Bonferroni method). For comparisons to ancestral λ, it was not possible to constrain comparisons to same-day measurements because ancestral λ was only measured on one day. To address this, we computed relative decay rates for each genotype by dividing the raw decay rate by the 6-mut decay rate measured on the same day. We then ran another set of comparisons between the ancestral λ and all other genotypes using the relative decay rates, and the same pairs were significant as with the raw decay rates, even with the stringent Bonferroni adjusted α value of 0.0016. Since the day effects were small compared to the differences between variants, we do not believe that the day-effects substantively affected our interpretation of the results. Both raw and relative decay rates are reported in S3 Table.

### Coevolution replay experiment

For the experiment including the naturally evolved stabilizing mutations and the engineered variant T987S:

On the day preceding the first day of the experiment, we induced phage from each genotype from their corresponding lysogens, as described in the section 'Induction of lysogens by heat shock'. We then filtered lysates through 0.22 $\mu M$ syringe filters and quantified lysate densities by serially diluting in the replay experiment media, M9 glucose, and spotting 2 $\mu L$ on a lawn of

WT cells. The next day, designated 'Day 1' of the replay, we inoculated 6 flasks per starting genotype with 9.7 mL M9glucose media, 100 $\mu L$ MgSO4, 100 $\mu L$ host bacteria, and phage. Each genotype was diluted to equalize the number of initial particles across genotypes. We added approximately $10^6$ phage to each initial flask, which corresponded to 50 $\mu L$ of 6-mut and 5 $\mu L$ of 6-mut T987A, 6-mut T987S, and 6-mut F1122L. Exact numbers of phage added were $1.8 \times 10^6$ for 6-mut, $4.9 \times 10^6$ for 6-mut T987A, $4.6 \times 10^6$ for 6-mut T987S, and $2.5 \times 10^6$ for 6-mut F1122L. For each phage genotype half of the replicates (3 out of 6 flasks) were initiated with REL606 wild type hosts and the other half were initiated with LRO1 hosts (a $malT^-$ derivative of REL606). Flasks were incubated for 24 hours at 37°C shaking. On day 1 of the experiment, we verified that no populations started the experiment as OmpF$^+$ by plating ~6x the volume of phage added to experimental flasks with $lamB^-$ bacteria. No OmpF$^+$ plaques were detected. Daily transfers were made (every 24 hours) from the previous day flask to a new flask with fresh M9 glucose and MgSO$_4$. Because the $malT^-$ is a more challenging host for phage to grow on compared to REL606, we transferred 10% of flask contents for $malT^-$ flasks and 1% of flask contents for REL606 flasks. After completing each daily transfer, we sampled 1 mL from the previous day's flask and processed the samples as follows. To each 1 mL sample we added 40 $\mu L$ chloroform and centrifuged at 3,214× g for 10 minutes. Then we spotted 2 $\mu L$ phage from each sample onto a lawn of $lamB^-$ cells to test for OmpF$^+$. We also serially diluted phage from each sample in M9 glucose and spotted 2 $\mu L$ of each dilution on a lawn of WT cells to estimate the total number of phage in each flask. To save samples for future analysis, we made a frozen stock of each population by adding 40 $\mu L$ 80% glycerol to 200 $\mu L$ sample. When OmpF$^+$ phage were detected on the $lamB^-$ plates, we isolated a single plaque and sequenced the reactive region of the J gene to identify mutations.

**For the experiments including the remaining engineered variants.** We measured evolvability of the engineered library variants using co-evolution replay experiments very similar to the one described in the previous section. Due to the number of variants in the library, we performed two separate 10-day experiments (experiment 1: 6-mut, T987L, T987G, T987A, T987C, T987Y; experiment 2: 6-mut, T987A, T987R, T987K). To gauge whether results of the two experiments could be directly compared, we included six replicates of 6-mut and six replicates of T987A in each experiment. In both cases the outcomes were identical or similar between experiments (6-mut: experiment 1: 6/6 replicates evolved, all six required 1 day, all six required 1 mutation; experiment 2: 6/6 replicates evolved, all six required 1 day, all required 1 mutation; T987A: experiment 1: 6/6 replicates evolved, average 4.2 days, all required 2 mutations, experiment 2: 4/6 replicates evolved, average 5.25 days, all required 2 mutations). The higher variation in T987A compared to 6-mut is expected given increased opportunities for stochasticity when two mutations are required instead of just one. This indicates that results are consistent and repeatable between experiments. For all variants in both experiments, between $9.6 \times 10^5$ and $3.0 \times 10^7$ phage were added to each experimental flask, consistent with the experiment that included naturally evolved thermostable genotypes. All other elements of the experimental design were kept as identical to the initial experiment as possible, with a few minor exceptions: (1) REL606 was used as the host in all six replicates, as opposed to the three MalT$^-$/ three REL606 design used for the initial experiment; (2) we carried out 1% transfers each day, consistent with the REL606 replicates from the initial experiment; (3) due to the larger size of the later experiment, we did not measure density of phage on WT each day, but instead spotted 5 $\mu L$ of undiluted chloroformed lysate from each flask on a WT lawn to determine the presence or absence of phage.

## Bioinformatic prediction of J structure

We used the publicly available version [49] of Alphafold [31] to predict the structure of a truncated version of J containing the 173 most C-terminal amino acids, for both the ancestor and the 6-mut. We used all recommended presets in the publicly available version, and sequence coverage and model confidence are shown in S2 Fig. We used Chimera [50] to visualize structures and create Figs 4 and S2.

## Reconstruction of evolutionary pathways from coevolution replay

When replay populations evolved OmpF[+], we sequenced the binding region of J from a single plaque on the first day OmpF[+] was detected. We chose to further explore one population from each background and reconstruct the genotypes that evolved by engineering the mutations into a lysogenic λ. The purpose of reconstructing genotypes of interest as lysogens was to (1) ensure stable long-term storage of these genotypes as lysogens and (2) ensure that subsequent decay assays would be performed on freshly produced phage particles generated by triggering lysis by heat shock rather than growing up overnight stocks, which may contain phage particles of varying ages. The oligos used to engineer these strains can be found in S2 Table. Decay assays were performed in the same way described in the section "**decay assays**". Due to the relatively low-throughput nature of our decay assay, we performed these decay assays in a blocked design, with three blocks measured on three different days. Each block consisted of the stabilized starting genotype, as well as derivatives that evolved during the experiment. In each block, we also measured 6-mut and 7-mut. Since 6-mut and 7-mut were measured on three different days, we were able to gauge how much variation in decay rates was due to our blocked design. We found that there were no significant differences among the decay rates for 6-mut measured on different days (one-way ANOVA, n = 3 per day per genotype, $F_{[2,6]}$ = 0.32, P = 0.7391), and the same was true for 7-mut (one-way ANOVA, n = 3 per day per genotype, F[2.6] = 4.12, P = 0.0748) indicating that there is very little variation across days. This allowed us to pool 6-mut and 7-mut decay rates across days, and we used a t-test on pooled 6-mut vs. pooled 7-mut (n = 9 per genotype) to measure the effect of the N1107K mutation in the 6-mut background. To compare among the genotypes that evolved in stabilized backgrounds (6-mut + stabilizing mutation; 6-mut + stabilizing mutation + putative destabilizing mutation; 6-mut + stabilizing mutation + putative destabilizing mutation + final gain-of-function mutation), we measured the five genotypes in each panel of Fig 5 on the same day and used paired t-tests to compare each genotype to its immediate ancestor, using only measurements taken on the same day. Significance was corrected for multiple testing using the Bonferroni method.

## Editing N1107K into all variant backgrounds

As an additional verification of our model, we engineered N1107K into all variants and measured OmpF use. For some backgrounds (6-mut, T987A, F1122L, T987L and T987C) we edited N1107K in using the same CoS-MAGE technique described in the section **"CoS MAGE to insert stabilizing mutations into 6-mut lysogen",** and then picked colonies and verified the change by sequencing. For the remaining engineered library variants (T987S, T987G, T987K, T987R, T987Y), we modified our CoS-MAGE protocol to increase throughput such that we could check whether N1107K confers OmpF[+] to each variant without picking colonies and sequencing individually for each variant (S1 Text). Regardless of whether the edited genotypes were produced in the low or high throughput method, we tested for OmpF use in edited genotypes first by plating dilution series of induced lysogens on *lamB*[−] lawns. As a second

measure of OmpF use, we also calculated growth rates on lamB⁻cells in liquid culture for a subset of variants, which allowed more sensitive detection of weak OmpF activity (S1 Text).

## Additional verification of plaque-based detection of OmpF⁺ with a liquid-based growth assay in populations from replay experiment

Because we observed that the 7-mut T987A and T987Y thermostable mutant plaqued poorly on *lamB*⁻cells yet grew measurably on *lamB*⁻cells in liquid culture, we wondered if any populations in the coevolution replay experiments had become OmpF⁺ without being detected by the plaque assay. To address this possibility, we measured growth on *lamB*⁻ in liquid among all six replicates of a subset of naturally evolved and engineered genotypes (6-mut, T987A, F1122L, and T987S) at day 6 of co-evolution. We combined 1 mL from each experimental flask with 40 $\mu L$ chloroform to kill the coevolving bacteria and centrifuged the tubes at 21,000× g for 1 minute. Then we inoculated 100 $\mu L$ from each population into culture tubes containing 4 mL LBM9, 40 $\mu L$ MgSO4, and 100 $\mu L$ *lamB*⁻cells. Tubes were incubated at 37°C shaking for 12.5 hours. After incubation, cells were treated with chloroform (40 $\mu L$ chloroform to 1 mL sample). Quantification of phage was assessed before and after incubation by serially diluting chloroformed phage in M9 glucose and spotting 2 $\mu L$ on a lawn of WT cells. The assay did not reveal any populations that grew on *lamB*⁻in liquid but were not detected by the plate-based assays. Growth rates are reported in S4 Table.

## Growth rate on REL606 in M9 glucose media

To assess the growth rate of each variant in the conditions used for the evolution experiment, we induced three replicates of each variant using the same procedure as described in the section **Induction of lysogens by heat shock,** filtered through a 0.22 $\mu M$ filter, and diluted each in M9 glucose + MgSO$_4$ depending on initial titer upon induction, such that ~10$^4$ phage were added to each flask containing 10 mL M9 glucose, 100 $\mu L$ MgSO4. Flasks were mixed well, and phage titers were measured by plating with WT cells infused in soft agar. Then, we added ~10$^8$ REL606 cells to each flask and incubated at 37 C shaking for 4 hours. After incubation, samples were taken from each flask, filtered through 0.22 $\mu M$ filters, and phage titers were re-measured by diluting in M9 glucose + MgSO$_4$ and plating with WT cells infused in soft agar. We chose to measure growth over 4 hours rather than 24 hours (the length of time between transfers in the evolution experiment) to reduce the possibility that genotypes would evolve mutations during the growth experiment. T987A, T987L, T987S, T987G, T987K, T987R, and T987Y were measured alongside 6-mut in a single experiment. F1122L and T987C were measured on a different day alongside three additional replicates of 6-mut, and no day effects were detected. Paired t-tests were used to compare growth rates of variants to 6-mut and corrected for multiple comparisons using the Bonferroni method ($\alpha$ = 0.0056).

## Assays for productivity in naturally evolved thermostable variants containing stabilizing mutations and N1107K

We used the same procedure as described in the section **"CoS MAGE to insert stabilizing mutations into 6-mut lysogen"** to edit N1107K into 6-mut and the two naturally evolved thermostable genotype (T987A, and F1122L). To test the productivity of these genotypes, we induced phage as described in the section 'Induction of lysogens by heat shock'. We then filtered lysates through 0.22 μM syringe filters, serially diluted in LBM9, and spotted 2 $\mu L$ of each dilution on a lawn of *WT* cells. Statistical comparisons were made between titers produced by

genotypes with and without N1107K using unequal variance t-tests and corrected for multiple comparisons using the Bonferroni method ($\alpha = 0.025$).

## Testing the effect of the T987A stabilizing mutation in 5-mut, a background further removed from OmpF+

We used the same procedure as described in the section **"CoS MAGE to insert stabilizing mutations into 6-mut lysogen"** to revert another one of the 7-mut mutations (changing the amino acid at codon 1012 from G back to S) to create a 5-mut. We then used CoS MAGE again to insert the stabilizing mutation T987A into the 5-mut. To assess whether T987A is stabilizing in this background, we used the same procedure described in the section "**Decay Assays**" to measure decay rates. The decay rates of 5-mut and 5-mut T987A were compared with an unequal variance 2-sample T-test. To assess evolvability, we used the same procedure described in the section "**Coevolution replay experiment**" to conduct a 10-day experiment with six replicate flasks each for 5-mut and 5-mut T987A. All six flasks were inoculated with an initial population of $10^6$ phage particles, and all six flasks contained the REL606 host bacteria. As with the other coevolution experiments, we transferred 1% of the flask contents (phage and host bacteria) every 24 hours and plated daily on WT and *LamB*- hosts.

## Software for statistics and plotting

All statistics and plotting were carried out in Matlab (version R2019b). Statistical methods are detailed in materials and methods subsections above. All t-tests were two-tailed. Data were checked for normality and homogeneity of variance prior to performing statistical tests.

## Dryad DOI

https://doi.org/10.6076/D1NC78 [51]

## Supporting information

**S1 Fig. Phage titer during evolution experiment on naturally occurring thermostable genotypes.** Phage titer measured on WT cells on each day of the evolution experiment on the naturally evolved thermostable variants (Fig 2).
(TIF)

**S2 Fig. Aligned Alphafold predictions for ancestor and 6-mut J proteins.** A: aligned Alphafold predictions for ancestor (purple) and 6-mut (teal) J protein reactive region (amino acids 960–1132). **B:** Coverage of multiple sequence alignment (MSA) used to make structural prediction and predicted IDDT (model confidence out of 100) at each position.
(TIF)

**S3 Fig. Editing N1107K into all variant backgrounds.** To verify that most of the stable genotypes required an additional mutation along with N1107K for OmpF+, we edited N1107K back into all genotypes in the library (S1 Text) and tested their ability to plaque on LamB−. N1107K conferred full OmpF+ on three genotypes (6-mut, T987L, and T987C), and these were all detected during the evolution experiment. Two variants, T987A and T987Y, were able to form very turbid clearings when spotted on LamB−, but because these genotypes were not able to form individual plaques we did not detect them during the evolution experiment. Consistent with their poor plaquing, genotypes with partial OmpF-use grew at a dramatically lower rate compared to genotypes with full OmpF-use (S4 Fig). All genotypes were spotted on WT lawns as a positive control to indicate that plaquing effect was specific to OmpF-use and not

indicative of generic viability.
(TIF)

**S4 Fig. Turbid clearings on plates correspond to weak growth in liquid. A:** Plate photo of each genotype spotted on a lawn of cells that express OmpF only.**B:** Growth rate of three replicates of each genotype on cells that express OmpF but not LamB (n = 3 for 6-mut N1107K and F1122L N1107K, n = 2 for T987A N1107K; paired t-tests corrected for multiple comparisons using Bonferroni method; 6-mut + N1107K vs. 6-mut T987A + N1107K: p = 6.38x10$^{-4}$; 6-mut + N1107K vs. 6-mut F1122L + N1107K: p = 3.38x10$^{-4}$; 6-mut + N1107K vs. 6-mut F1122L + N1107K: p = 1.06x10$^{-6}$. Bonferroni adjusted significance: ns: > 0.0167, *: p < 0.00167, **: p < 1.67e$^{-4}$, *** p < 1.67e$^{-5}$, ****: p < 1.67e$^{-6}$.)
(TIF)

**S5 Fig. Independently created T987C lysogen.** To rule out the possibility that T987C had acquired mutations outside of J that helped it evolve OmpF$^+$ faster, we re-engineered a new lysogen with an oligo specifically designed to produce T987C. We then measured its evolvability in 12 replicate populations, as well as 12 replicate populations of the 6-mut as a control. All 12 T987C replicate populations evolved OmpF$^+$ in one day, compared to eleven of twelve 6-mut replicate populations, confirming that these genotypes have nearly identical evolvabilities under the conditions used in our evolution experiments.
(TIF)

**S6 Fig. Productivity of stable genotypes with and without N1107K.** Productivity (log viable titer) of each genotype immediately after lysogen induction, with and without N1107K mutation. Statistical comparisons were made using two t-tests and corrected for multiple comparisons using the Bonferroni method, N = 3 per genotype. 6-mut T987A vs. 6-mut T987A N1107K: p = 0.4376, 6-mut F1122L vs. 6-mut F1122L N1107K: p = 0.3508. Bonferroni adjusted significance: ns: p > 0.025.
(TIF)

**S1 Table. High throughput MAGE to engineer N1107K into genotypes that either did not evolve OmpF$^+$ in the evolution experiment (T987G) or required two mutations to evolve OmpF$^+$ (T987S, T987K, T987R, T987Y).** 6-mut was included as a positive control because it is expected to convert to OmpF$^+$ upon receiving the N1107K mutation. As an additional positive control to ensure that the MAGE process was working for each variant, we used an oligo mix containing N1107K as well as the appropriate *galK* conversion oligo and computed the fraction of cells that were successfully converted from *galK*$^+$ to *galK*$^—$ or vice versa. We computed the conversion rate to OmpF$^+$ after MAGE by inducing the lysogens and plating dilutions of each lysate on two lawns: one with cells expressing only OmpF, and another with WT cells. We then divided the number of plaques on the OmpF only lawn by the total number of plaques on the WT lawn to get the conversion rate. In T987S, T987K, T987R, and T987Y we did not observe any OmpF$^+$ plaques, so we computed an upper bound as the conversion rate if a single OmpF$^+$ plaque had been observed. For T987A, F1122L, T987L and T987C, we separately engineered in N1107K and isolated and sequenced individual clones rather than using the high throughput method described here.
(DOCX)

**S2 Table. MAGE oligos.** Asterisks indicate phosphorothioated bond and underlined letters indicate the mutated bases.
(DOCX)

**S3 Table. Unadjusted and adjusted decay rates for engineered library variants.**
(DOCX)

**S4 Table. λ growth rates on *lamB⁻* hosts measured in liquid culture on D6 of the replay experiment. Positive growth rates are bold.** Growth rates could not be calculated ("NA") for some populations (6-mut T987A 2 and 6-mut F1122L 1 and 2) because no viable phage were detected at day 6 (i.e. the population had gone extinct), or because phage decayed to zero during the overnight growth period on *lamB⁻* (6-mut T987S 4). Growth rates for each population were measured in a single replicate.
(DOCX)

**S1 Text. Text on two additional experiments.** First, a description of a high throughput genetic engineering method and assay for evaluating whether lambda strains are one mutation (N1107K) away from gaining the use of OmpF. Second, a description of a more sensitive assay for detecting OmpF use than plate-based spot tests.
(DOCX)

## Acknowledgments

We would like to thank Sarah Medina for help in the laboratory. We would also like to thank Katie Petrie, Morgan Mouchka, Animesh Gupta, and Josh Borin for valuable discussion of the manuscript.

## Author Contributions

**Conceptualization:** Hannah M. Strobel, Justin R. Meyer.

**Funding acquisition:** Justin R. Meyer.

**Investigation:** Hannah M. Strobel, Elijah K. Horwitz, Justin R. Meyer.

**Methodology:** Hannah M. Strobel, Elijah K. Horwitz, Justin R. Meyer.

**Writing – original draft:** Hannah M. Strobel, Justin R. Meyer.

**Writing – review & editing:** Hannah M. Strobel, Justin R. Meyer.

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
