## [Decision Letter · Decision Letter 0]

29 Oct 2021

Dear Dr Meyer,

Thank you very much for submitting your Research Article entitled 'Viral protein instability enhances emergence evolvability' to PLOS Genetics.

The manuscript was fully evaluated at the editorial level and by independent peer reviewers. The reviewers and editorial team were enthusiastic about this contribution, but the reviews raised several concerns about the current manuscript. Based on the reviews, we will not be able to accept this version of the manuscript, but we would be willing to review a revised version. We cannot, of course, promise publication at that time.

In particular, it is critical that the revision address the fact that these results could be seen as self-evident ("unsurprising"), given that key mutations required for the use of OmpF are already known to be destabilizing.  Closely related to this issue is the question of the extent to which these results are generalizable, rather than specific to this particular phage-host system.  The latter concern should be carefully discussed and some of the language that potentially over-generalizes the results (to the emergence of animal viruses) should be revisited.  The reviewers also had concerns about the limitations of both the experimental and modelling approaches which should be carefully addressed.

If you decide to revise the manuscript for further consideration at PLOS Genetics, please aim to resubmit within the next 60 days, unless it will take extra time to address the concerns of the reviewers, in which case we would appreciate an expected resubmission date by email to plosgenetics@plos.org.

[LINK]

We are sorry that we cannot be more positive about your manuscript at this stage. Please do not hesitate to contact us if you have any concerns or questions.

Yours sincerely,

Lindi Wahl

Associate Editor

PLOS Genetics

Kirsten Bomblies

Section Editor: Evolution

PLOS Genetics

Reviewer's Responses to Questions

**Comments to the Authors:**

Reviewer #1: In this paper, Strobel et al. attempt to examine the link between thermostability and evolvability. They use bacteriophage lambda which they have previously shown can evolve from using only LamB as a receptor to also being able to use OmpF as a receptor. Previously, they found that an isolate with 7 mutations (7-mut) had the ability to use LamB and OmpF but was much less stable than the ancestor. They reversed a key mutation for OmpF use and found that the resulting isolate (6-mut) was still unstable. They then introduce two different stabilizing mutations to 6-mut. Following experimental evolution, all populations of 6-mut rapidly gained the ability to use OmpF whereas the stabilized populations took much longer, and many populations did not evolve the ability or went extinct. Sequencing revealed individual mutations gave 6-mut the ability to grow on OmpF but each stabilized population required 2 separate mutations to evolve to grow on OmpF. Next, they mutate one of the residues which increased thermotolerance to generate a panel of mutants. These mutants differed in thermostability and following experimental evolution, the more thermostable mutants evolved slower with a single exception (T987C). Again, thermostable isolates required 2 mutations.

Next, the authors use Alphafold to model the predicted structure of the viral protein J, the host recognition protein, to show the predicted location of the mutations from their experiments. This showed that mutations allowing the use of OmpF were in sensible places on an exposed surface of the protein. They then show that the double mutations required for the thermostable variants to evolve to use OmpF, the first mutation destabilizes the phage before the second allows the use of OmpF. Introducing, only the mutation allowing the use of OmpF to thermostable isolates did not allow robust use of OmpF without an additional mutation. Finally, they measure growth rates of the variants and show that T987C has significantly lower growth compared to all the other isolates which explains why it might not evolve naturally. The authors propose a model that J must be destabilized such that it can access alternative forms of the protein which can bind to OmpF.

The paper is clearly written, and the experiments are easy to follow. The authors have generated some interesting data and this paper forms part of an important debate on the relationship between thermostability and evolvability. I especially like the model that the authors propose that viral attachment proteins must be unstable to infect a broader range of hosts. However, I think the authors should address 3 major things before publication.

1. The authors don’t discuss the limitation of their experiment with 6-mut being a single mutation away and 1122 being effectively a reversion of a previous mutation.

2. The modelling section needs more discussion of the model generated and the limitations of this approach.

3. I would like more discussion of the limitations of the experimental design and more links to the published literature.

Major Issues

1. I have a conceptual issue with the paper. The authors start with an isolate which is a single mutation away from 7-mut, the isolate with ability to use OmpF. Unsurprisingly, OmpF use can evolve very quickly as there were a large number of phage in the initial population. This was compared to isolates with additional stabilizing mutations. However, the authors in their previous work have already shown that the key mutations allowing the use of OmpF are destabilizing. When tracking the mutations, the stabilizing mutation reverted before the OmpF allowing mutation arose. In fact, 1122 was the site of one of the mutations in 6-mut so using it as a thermostability mutation is in effect creating a 5-mut. It is not then surprising that the thermostable mutants took longer to evolve on OmpF. Is this therefore a good/fair test of evolvability? Is 1122, the key mutation that destabilizes 6-mut? The authors are in effect taking something where it is known that the required mutations are destabilizing, reversing those mutations and showing that it takes longer to evolve the trait. There was no way that anything could evolve faster than 6-mut as it required just a single mutation which happened within a passage or two. It seems likely that for epistatic reasons, a more stable J cannot accept the OmpF-using mutations and therefore required 2 mutations.

All this being said, even if the experimental design is not perfect, I still believe that this data is significant, and it highlights that for the phage to switch receptors, it requires a destabilization of the protein. I think the authors should discuss whether this is a special case of evolvability and discuss the caveats of their approach in much greater detail.

2. The Alphafold modelling section needs to be reworked. Firstly, Alphafold provides a confidence in its own model prediction. This data needs to be shown so that we can see whether the conclusions drawn from the model are likely. I would guess that the 960-990 region is low confidence which would not be surprising because it could likely interact with the N-terminal portion of the protein. The decision to model only 173 amino acids rather than the full protein should be discussed not just in the methods. Maybe a schematic should be provided of the known domains of J. Alphafold models exist of longer proteins than J so the authors could try to model the full structure. The accuracy of the Alphafold prediction is greatly improved by having a good Multiple Sequence Alignment (MSA) of related proteins. What was the depth of the MSA, how did this affect the confidence in the prediction? Are there any related structures which have been experimentally determined that this structure could be compared to? Finally, the authors could consider trying to model how their structure might interact with LamB or OmpF although this may be beyond the scope of the paper.

3. I would like to see more discussion especially of the limitations of the paper and comparing to other research. For example, mutations at a single site are unlikely to have independent effects from each other but the authors don’t mention this as a pretty big limitation of their experimental design.

What are the effects of performing this evolution experiment in laboratory conditions? The authors by passaging regularly, are removing any cost of a lack of environmental survival. Does a lab environment allow unstable phage that the real world does not? This should be discussed.

I think that the authors should engage more with the literature about why their results differ from the consensus. I think the current model suggests that evolvability can be increased by having a stable core which allows an active site to change without destroying the protein fold. This paradigm has mostly been applied to globular enzymes. This paradigm might not apply to bacteriophage attachment proteins. This paper suggests to me that to evolve to use a new receptor, J must be more flexible which in turn reduces the stability of the phage. There is research on bacteriophage life history trade-offs that discusses this hypothesis.

More examples of different viruses evolving would also benefit the discussion.

Minor Issues

1. Figure 3A should be presented the other way up. It’s easier if there is consistency between the figures so higher values = more stable.

2. Please add Figure 6 stats.

3. L40. I think the authors can assert that viruses are highly evolvable.

4. L59. 2019 rather than 2020 as the start of the pandemic.

5. L88 non-native conformers. An additional sentence explaining what this means would be helpful as I was not familiar with this phrase.

6. L94. Did the authors mean apposing here?

7. L138 and elsewhere. The authors perform multiple T tests on their data where I think that a 1-way ANOVA correcting for multiple comparisons would be more appropriate and is what the authors use later.

8. L152- State that the coevolved host is LamB-.

9. L173-L175. Should state that L1122F is a direct reversion. These sentences could be better integrated into the paragraph. Rephrase to make a little clearer.

10. L204- No need to give the base changes. Amino acid changes only would be clearer.

11. Supplemental figure S3. This figure is not supplemental in my opinion but is key to understanding the entire mechanism proposed. This data demonstrates that N1107K does not give the same effect in a stabilized J protein which is why 2 mutations are required. I think it (or a quantification of the results) should be moved to a main figure.

12. L310-311 Not clear when and where the authors expected this mutation to evolve.

13. L317 Kruskal Wallis or ANOVA? Which one? Why was Kruskal Wallis used later on but ANOVA before?

14. L410-414. This is quite speculative. I can see links between this work and the propensity for furin cleavage in S and up/down conformations. The authors could discuss this in more detail though not if they don’t want to. Given how variants have outcompeted the initial strain, I don’t know how great an example this is for the paper even if topical.

Reviewer #2: I understand that the authors want to publicize their work by referring to viral emergence and the COVID pandemics. However, this article deals with a very specific phage-bacterial system, and extrapolations to the use of receptors in animal viruses are not warranted in this case. Therefore, I suggest refocusing these references (last paragraph, author's abstract). Host shifts are also relevant in the bacterial world, for example for phage therapy. I suggest that the word "emergence" be removed from the title.

The authors measured overall infectious particle stability (decay assays), not the stability of a given protein. This should be OK since mutants were isogenic, but some caveats apply though.

Since the experiments are based on modifying thermostability by changing one or a few sites, the question arises whether the observed differences in evolvability are due to specific epistatic interactions between sites apparently affecting stability and gain-of-function mutations. The authors select a site (987) that is apparently physically separate from the region that determines ompF evolution. However, this does not exclude these possible "confounding effects", as interactions between the sites could occur in any case.

Finally, the relevance and generality of the conclusions is affected by the fact that, in the model system, the evolutionary trajectory (acquisition of ompF) is rather short (one or few mutations). No other phenotype was examined.

**Have all data underlying the figures and results presented in the manuscript been provided?**

Reviewer #1: Yes

Reviewer #2: None

PLOS authors have the option to publish the peer review history of their article (what does this mean?). If published, this will include your full peer review and any attached files.

Reviewer #1: No

Reviewer #2: No

---

## [Decision Letter · Decision Letter 1]

4 Jan 2022

Dear Dr Meyer,

Thank you very much for submitting your Research Article entitled 'Viral protein instability enhances host-range evolvability' to PLOS Genetics.

The manuscript was fully evaluated at the editorial level and by independent peer reviewers. Both reviewers appreciated your care and thoroughness in addressing their previous concerns.  The reviewers have suggested a few more clarifications, all relatively minor, that we ask you to include in a revised manuscript.

We therefore ask you to modify the manuscript according to the review recommendations. Your revisions should address the specific points made by each reviewer.  These consist of a couple clarifications and/or caveats that each reviewer has requested you add to the text.  All of these suggestions should be taken on board, with particular attention to mutations at the 1122 site (reviewer 1) and to the generalizability (or not) of these results (reviewer 2).

[LINK]

Yours sincerely,

Lindi Wahl

Associate Editor

PLOS Genetics

Kirsten Bomblies

Section Editor: Evolution

PLOS Genetics

Reviewer's Responses to Questions

**Comments to the Authors:**

Reviewer #1: Please see attachment.

Reviewer #2: I understand that the authors want to publicize their work by referring to viral emergence and the COVID pandemics. However, this article deals with a very specific phagebacterial system, and extrapolations to the use of receptors in animal viruses are not warranted in this case. Therefore, I suggest refocusing these references (last paragraph, author's abstract). Host shifts are also relevant in the bacterial world, for example for phage therapy. I suggest that the word "emergence" be removed from the title.

We have changed the word “emergence” to “host-range” in the title and altered the abstract, author summary, and to more generally discuss virus host-range evolution. However, we would like to push back on the idea that bacteriophages provide no useful insights into animal virus evolution. Leaders in the field of experimental evolution routinely use bacteriophages as model systems to broadly understand the evolution of viruses, and previous papers have discussed how their results provide insights into the evolution of medically relevant viruses (two highly cited examples are listed below). With that in mind, we added a sentence in our final paragraph to make our conclusions more conservative with respect to animal virus applications. “Our results were derived from a study of bacteria-infecting virus rather than an animal virus, and further work is warranted to investigate the role of viral particle instability in animal virus host-range evolvability.” And we also add specific recommendations for how our findings might be applied in this context.

R. C. McBride, C. B. Ogbunugafor, P. E. Turner, Robustness promotes evolvability of thermotolerance in an RNA virus. BMC Evolutionary biology 8: 231 (2008). C. B. Ogbunugafor, R. C. McBride, P. E. Turner. Predicting virus evolution: the relationship between genetic robustness and evolvability of thermotolerance. Cold Spring Harbor Symposia on Quantitative Biology LXXIV, 109-118 (2009).

J. M. Cuevas, A. Moya, R. Sanjuán. A genetic background with low mutational robustness is associated with increased adaptability to a novel host in an RNA virus. J Evol Biol 22, 2041-2048 (2009).

-> OK. Please notice, though, that only the McBride paper was conducted with phages.

The authors measured overall infectious particle stability (decay assays), not the stability of a given protein. This should be OK since mutants were isogenic, but some caveats apply though.

We acknowledge the reviewer’s hesitance to accept the particle stability as a proxy for the stability of the J protein. We have several pieces of evidence to support our claim. The first is that, as the reviewer recognized above, the mutants were isogenic. And additionally, they were engineered using oligos with the specific mutation, so it would be highly surprising if the stabilizing effect of all of the mutants were due to mutations appearing elsewhere in the genomes in parallel, despite no selection for particle stability when the phage are in the lysogenic state. The second piece of evidence is the subject of a project that is currently underway in the lab, and we anticipate will be published in the next two years. We have successfully purified active J protein from ancestral and OmpF+ mutants and found that the levels of soluble expression and melting points of the OmpF+ J protein are dramatically lower than the OmpFancestor. While the OmpF+ J proteins we have studied have a slightly different set of mutations than the unstable OmpF+ genotypes studied in this paper, we see this as a promising sign that our measurements on whole phage stability are reflective of differences in J protein stability. These results are beyond the scope of the current paper, but we are eager to prepare this work for publication soon.

-> OK, this is convincing

Since the experiments are based on modifying thermostability by changing one or a few sites, the question arises whether the observed differences in evolvability are due to specific epistatic interactions between sites apparently affecting stability and gain-of-function mutations. The authors select a site (987) that is apparently physically separate from the region that determines ompF evolution. However, this does not exclude these possible "confounding effects", as interactions between the sites could occur in any case.

We do not rule out the possibility that there are specific interactions between the site 987 and the region that determines OmpF evolution, and it is possible (likely even) that stabilizing different regions of the protein would result in different effects on evolvability. As to whether the loss in evolvability is only due to epistasis between stabilizing and gain of function mutations, we address this with additional experiments reported in the manuscript: “Along similar lines of thinking, perhaps the extra mutation is required to ameliorate a genetic incompatibility between the stabilizing mutations and the mutations found in the contact surface. This could occur if stabilizing mutations and surface mutations are beneficial alone but produce a nonfunctional protein when they co-occur in the same genotype. To rule out this possibility, we measured productivity of lysogens for the naturally selected thermostable variants (T987A, F1122L) with and without N1107K and found no difference in titer (Fig. S6) indicating that there is not a genetic incompatibility [33]. “

-> Seems good.

Finally, the relevance and generality of the conclusions is affected by the fact that, in the model system, the evolutionary trajectory (acquisition of ompF) is rather short (one or few mutations). No other phenotype was examined.

We conducted additional experiments in response to this concern. To test the effect of a stabilizing mutation in a background that is further away from the OmpF+ 7-mut, we started with the 6-mut and reverted another mutation, creating a 5-mut. We then edited in the T987A stabilizing mutation, confirmed that it is stabilizing in the 5-mut background, and measured its evolvability relative to 5-mut. We found that, completely in alignment with our model, the stabilizing mutation inhibits evolvability in the 5-mut background. In fact, the difference in evolvability is even greater than that between 6-mut and 6-mut T987A. These results are reported in the last paragraph in the results and Fig 7.

-> This a valuable experiment, but it doesn´t fully address my concern since it focuses on the same trait and evolutionary trajectory. I recommend that the authors add a comment on this to at least acknowledge that the results might potentially be somewhat determined by the specific mechanistic details at play, or may reflect local features of a fitness landscape, rather than reflecting a general process.

**Have all data underlying the figures and results presented in the manuscript been provided?**

Reviewer #1: Yes

Reviewer #2: None

PLOS authors have the option to publish the peer review history of their article (what does this mean?). If published, this will include your full peer review and any attached files.

Reviewer #1: No

Reviewer #2: No

---

## [Editor Report · Decision Letter 2]

11 Jan 2022

Dear Dr Meyer,

We are pleased to inform you that your manuscript entitled "Viral protein instability enhances host-range evolvability" has been editorially accepted for publication in PLOS Genetics. Thanks for your careful work in addressing the revisions and congratulations on this excellent contribution to the field.

Yours sincerely,

Lindi Wahl

Associate Editor

PLOS Genetics

Kirsten Bomblies

Section Editor: Evolution

PLOS Genetics

**Data Deposition**

http://datadryad.org/submit?journalID=pgenetics&manu=PGENETICS-D-21-01290R2

**Press Queries**

---

## [Editor Report · Acceptance letter]

13 Feb 2022

PGENETICS-D-21-01290R2 

Viral protein instability enhances host-range evolvability 

Dear Dr Meyer, 

We are pleased to inform you that your manuscript entitled "Viral protein instability enhances host-range evolvability" has been formally accepted for publication in PLOS Genetics! Your manuscript is now with our production department and you will be notified of the publication date in due course.

With kind regards,

Zsofia Freund

PLOS Genetics

On behalf of:
